

# Mesoscale cascades and the "conundrum" of energy transfer from large to dissipation scales in an adiabatic ocean

Mikhail S.Dubovikov

NASA, Goddard Institute for Space Studies, 2880 Broadway, New York, NY, 10025,
Center for Climate Systems Research, Columbia Univ., New York, NY, 10025

Keywords: Energy conversion, Kelvin-Helmholtz instability, Stratified turbulence.

Author: mikhail.dubovikov@nasa.gov, **m.dubovikov@gmail.com,** tel.: 212-678-5608



## Abstract

A well-known "conundrum" in ocean dynamics has been expressed as follows: *"How does the energy of the general circulation cascade from the large climate scales, where most of it is generated, to the small scales, where all of it is dissipated? In particular, how is the dynamical transition made from an anisotropic, 2D-like, geostrophic cascade at large scales-with its strong inhibition of down-scale energy flux-to 3D-like, down-scale cascades at small scales."* (Muller, McWilliams and Molemaker, 2002). To study this as yet unsolved problem, we introduce in the analysis a dynamical consideration based on the mesoscale model developed by Dubovikov (2003) and Canuto and Dubovikov (2005) within which in a quasi-adiabatic ocean interior the large scale baroclinic instability generates mesoscale eddy potential energy (EPE) at scales of the Rossby deformation radius $\sim r_d$. Since at those scales the mesoscale Rossby number is small, the generated EPE cannot convert into eddy kinetic energy (EKE) and cascades to smaller scales at which the spectral Rossby number $Ro(k)$ increases until at some horizontal scales $\sim \ell$ it reaches $Ro(1/\ell) \sim 1$. Under this condition, EPE converts into EKE and thus the cascade of the former terminates while the inverse EKE cascade begins. At scales $\sim r_d$ the inverse EKE cascade terminates and reinforces the EPE cascade produced by the large scale baroclinic instability thus closing the mesoscale energy cycle. If the flow were exactly adiabatic, i.e. eddy energy were not dissipated, the latter would increase unlimitedly at the expense of the permanent production of the total eddy energy (TEE) by the mean flow. However, at the same scales $\sim \ell$ where the EPE cascade terminates and the inverse EKE cascade begins, the vertical eddy shear reaches the value of the buoyancy frequency $N$ that gives rise to the Kelvin-Helmholtz instability. The latter generates the stratified turbulence which finally dissipates EKE. A steady state regime sets in when the dissipation balances the TEE production by the mean flow.



## 1.Introduction

For a long time it is recognized that one of the most enigmatic conundrums of ocean general circulation is: "how and where does the energy of the general circulation cascade from the large climatic scales, where most of it is generated, to the small scales, where all of it is dissipated?" "In particular, how is the dynamical transition made from a 2-D like geostrophic cascade at large scales-with its strong inhibition of down-scale energy flux-to the more isotropic, 3D-like, down-scale cascades at small scales" (Muller et al., 2002; Mc Williams, 2003). In fact, interior ocean flows in the adiabatic approximation are sets of 2D ones within isopycnal surfaces between which non-linear interactions vanish. As discussed by Kraichnan (1965, 1967), in 2D flows only inversed (upscale) energy cascades are possible. Dubovikov (2003, D3) and Canuto and Dubovikov (2005, CD5) argued that mesoscale eddy energy (EKE) does cascade upscale, a process that is now commonly recognized (Ferrari and Wunsch, 2009; Bruggemann and Eden, 2015; Jansen et al., 2015) and confirmed by sea surface height (SSH) data (Scott and Wang, 2005; Scott and Arbic, 2007). It is worth noticing that the mentioned energy cascade is rather different than that in the two-level geostrophic model (Salmon, 1978,1980, 1998) within which at scales larger than the Rossby deformation radius $r_d$ there are the down-scale baroclinic energy cascade and the upscale barotropic one while at scales less than $r_d$ both cascades are directed down-scale. It is clear that such a flow contains no "conundrum". It is widely thought (e.g. Bruggemann and Eden, 2015) that because of the inverse direction of the energy cascade, the EKE has to be dissipated at basin scales. Several such mechanisms have been discussed: bottom friction (Arbic et al., 2009; Wunsch and Ferrari, 2004; Muller et al., 2002; Barkan et al., 2015), energy dissipation in the vicinity of the surface (Muller et al., 2002; Ferrari and Wunsch, 2009), generation of internal lee waves (Nikurashin and Ferrari, 2011; Muller et al., 2002). In particular, Nikurashin and Ferrari (2011) estimated the dissipation through the lee waves to be 0.2 TW and the bottom drag 0.12 TW which is somewhat smaller than the results of fine resolution models for bottom friction by Arbic et al. (2009) ranging between 0.14 and 0.65 TW as well as by Bruggemann and Eden (2015), $0.31\pm$ 0.23 TW. These results should be compared with the 1 TW wind power input into the ocean as estimated by Wunsch (1998) and 1.85 TW as estimated by von Storch (2012) from eddy-permitting simulations. Wunsch and Ferrari (2004) and Ferrari and Wunsch (2009) concluded "that drag in the bottom boundary layer is too weak to represent the dominant eddy energy sink". The opposite opinion is expressed by Jansen et al. (2015): "there is both observation and numerical evidence for strongly enhanced dissipation near the bottom boundary". Nevertheless, they recognized that "exact pathways of mesoscale EKE to dissipation remain unknown" what is in accordance with the conclusion of Wunsch and Ferrari (2004): "little is known about mesoscale eddy dissipation; models can say little". To clarify this issue in numerical models, one needs to perform simulations resolving the interval from the large scales ~1000 km to the Kolmogorov ones that is impossible now. Thus, at present the problem can be studied in analytical models only like the mesoscale dynamical model D3/CD5,6. Within that model, in the propounded essay we propose a pathway to the dissipation of mesoscale energy and a solution of the outlined conundrum together with the other mesoscale problems (we restrict ourselves by considering an adiabatic ocean only and, therefore, don't account for effects of the surface and boundary layers like bottom drag). In the development of the proposed pathway we apply the following inputs: (1) the EKE cascade in 2D flows is always inverse (upscale), (2) the EPE cascade is always down-scale, (3) intense release EPE$\rightarrow$EKE begins at scales where the spectral Rossby number $Ro(k)$ which at large scale is small, increases to unity, (4) generation of the stratified turbulence (ST) begins at scales where the





spectral Richardson number $Ri(k)$ which at large scales is large, decreases to unity and thus the Kelvin-Helmholtz instability occurs, (5) at scale $\sim r_d$ the mean potential energy (MPE) generates EPE only, (6) at scale $\sim r_d$ there occurs the transition EKE→EPE but not EPE→EKE, (7) the assumption that in the scale interval between one kilometer and several tens meters which thus far was not resolved in fine resolution simulations of mesoscales, the slopes $m$ of the EKE and EPE spectra are approximately the same $m \sim 2$ as at resolved scales from one kilometer up to $r_d$. The effects (1)-(5) are commonly recognized. The effect (6) follows from the D3/CD5 model which is based on equations for mesoscale fields at scales $\sim r_d$ with account for non-linear terms by contrast to the linear approximation widely used for study of mesoscales beginning with the Eady model. In order to check the assumption (7), it is necessary a better resolution than achieved thus far. Still, the assumption looks rather realistic since the interval between one kilometers and, say, 100 meters is less than that between $\sim r_d$ and one kilometer. Before applying the D3/CD5 model for non-linear terms to mesoscales, we developed and successfully tested a model of non-linear terms in different turbulent flows as presented in a series of paper (Canuto and Dubovikov, 1996-1999). Modeling of non-linear terms in D3/CD5 mesoscale equations allowed us to obtain a set of quantitative results compared favorably with observation and OGCMs data the majority of which are crucially determined by the non-linear interactions. The set of favorable comparisons of the model predictions with observation data is presented by Canuto et al. (2017): a) WOCE (2002) data for mesoscale kinetic enrgy in different locations, b) Philips and Rintoul (2000) measurements data for mesoscale diffusivity in ACC (143E, 51S), c) global data for mesoscale drift velocity by Fu (2009) and Chelton and Schlax (2011) at 150W and 110 W. In fine resolution simulations Luneva et al. (2015) validated the model parameterizations for the surface eddy kinetic energy and for the vertical buoyancy flux in the mixed layer. It is worth mention the validation of the model parameterization for the sub-mesoscale tracer flux with account for the effect of wind by Canuto and Dubovikov (2010) with use of fine resolution simulation data by Capet et al. (2008).

In the present study we discuss the following problems:

(I) At what scales does the inverse energy cascade start up and what is its source?

(II) Why does the kinetic energy spectrum that begins at the large climatic scales ~1000 km where the most part of the energy is generated, have a maximum not in the vicinity of the upper boundary of the spectrum like in 3D turbulent flows, but at mesoscales $\sim r_d$ where $r_d$ is the Rossby deformation radius?

(III) What is the sink of the inverse (upscale) EKE cascade? What energy reservoir does the cascade flow into: either (a) mean kinetic energy (MKE) or (b) eddy potential energy (EPE)? Results of studies within different approaches reject scenario (a) showing a rather weak exchange between EKE and MKE (see, for example, results of the numerical simulations by Boning and Budich, 1992, Figs. 8,9, and of the ocean analog of the observed atmospheric Lorenz energy cycle, 1960, summarized by Holton, 1992, Fig.10.13 adapted from Oort and Peixoto, 1974). The same conclusion follows from the analytical mesoscale model D3, CD5 referred to above (see Appendix). As for as scenario (b), at first sight it is inconsistent with the just cited data which, as expected, show the opposite transformation EPE→EKE after integrating over the whole spectra. However, this does not mean that EPE releases in all regions of its spectrum. Indeed, if at scales $\sim r_d$ the release EPE→EKE occurred as well, EKE would be unlimitedly stored at dispense of the inverse EKE cascade due to the weakness of the EKE→MKE transfer (see Appendix). Thus, we may expect that at scales $\sim r_d$ the transfer EKE→EPE occurs. In addition, only a down-scale



cascade of EPE could prevent its unlimited store at those scales. Note that the down-scale direction of cascades of variances of eddy scalar functions is the inherent property of 2-D flows (Lesieur, 1990). Within D3, CD5 model the discussed condition is satisfied.

As for the problem (I) about scales $\ell$ where the inverse energy cascade starts due to the release EPE→EKE, here we have no such quantitative model as at scales $\sim r_d$. Nevertheless, we will be able to carry out a semi-quantitative analysis to evaluate scales $\ell$ where the spectral Rossby number $Ro(k)$ reaches unity, i.e. $Ro(1/\ell) \sim 1$. Under this condition EPE releases and generates the inverse cascade of EKE. This occurs at $\ell \sim 100\mathrm{m}$. To perform the analysis, we assume that the slopes of the EKE and EPE spectra are similar to ones found at scales of few kilometers in fine resolution simulations. Furthermore, at the same scales $\sim \ell$ the spectral Richardson number decreases to unity that allows the vertical shear fluctuations to create the Kelvin-Helmholtz instability. The later generates the stratified turbulence (ST) which produces a down-scale 3D energy cascade to the Kolmogorov scales where the energy is dissipated. Thus, at scales $\sim \ell$ there is a competition between the production of EKE which cascades upscale, and that of ST which dissipates the eddy energy. At scales $\sim r_d$ the EKE cascade terminates and together with the baroclinic instability generates the EPE cascade closing the mesoscale energy cycle as presented in Fig.1. It is clear that if the generation of the inverse EKE cascade exceeds that of ST one, the mesoscale energy is dissipated weakly and thus the maxima of the EKE and EPE spectra at scales $\sim r_d$ become rather pronounced. And vice versa, if the generation of ST exceeds feeding EKE cascade, the latter is weak so that the maximum of the EKE spectrum cannot be so pronounced. Since numerous observations in the real Ocean show a considerable growth of the EKE spectrum at scales $\sim r_d$ (Stammer, 1998), we may conclude that at scales $\sim \ell$ the converse of the down-scale EPE cascade into the up-scale EKE one exceeds the generation of ST that is in accordance with the quasi-adiabatic nature of the interior ocean flow. The outlined conversion EKE $\rightleftarrows$ EPE is analogous to the conversion KE $\rightleftarrows$ PE in the oscillation of a pendulum with a low attenuation. In steady state the loss of EKE due to the generation of ST is balanced by feeding the total eddy energy (TEE=EPE+EKE) by the mean flow. The outlined scenario shown In Fig.1 which we substantiate below, suggests a solution of the conundrums listed above and details the internal structure of the eddy blocks in the schemes of the energy exchange between large scale and mesoscales in the ocean interior (see, for example, Boning and Budich, 1992, Figs. 8,9 and the oceanic analog of the observed atmospheric Lorenz energy cycle summarized by Holton, 1992, Fig.10.13). In Fig.1 we omit the block MKE since, on the one hand, the exchange between MKE and EKE is small in comparison with the exchange between EPE and MPE, as we show in Appendix and have noticed above, and, on the other hand, in the present study we are not interested in the energy exchange between MKE and MPE.

The organization of the paper is as follows. In section 2 we discuss the budget equations for the EKE and EPE. In section 3 we outlines the EPE and EKE mesoscale cascades and their mutual conversion in the framework of the idealized scheme when the conversion EKE→EPE occurs at scales $\sim r_d$ while EPE→EKE at the minimal scales of the cascades $\sim \ell$. In section 4 we list conditions of realization of the sketched scenario which we discuss in more details in the rest part of the paper. In particular, in section 5 within the D3 and CD5 mesoscale models we evaluate the transition EKE→EPE at scales $\sim r_d$ while in section 6 the transition EPE→EKE at scales $\sim \ell$. In section 7 we discuss the loss of TEE due to the production of ST and show that it occurs at the same scales $\sim \ell$ where there are the termination of the EPE cascade and the beginning of



EKE one. In section 8 we evaluate the vertical scale of the generated ST turbulence and in section 9 we summarize the obtained results. In Appendix we evaluate the energy exchange between MKE and EKE and show that it is weaker than the generation of EPE by MPE.

## 2. Production of EKE and EPE in the adiabatic limit

We begin with the equations for the horizontal eddy velocity and buoyancy fields which are obtained from the equations for the full fields $\mathbf{u} = \overline{\mathbf{u}} + \mathbf{u}'$ and $b = \overline{b} + b'$ by subtracting equations for averaged fields (bar denotes large scale fields averaged over sub-grid scales somewhat exceeding the eddy ones while prime marks eddy fields). In adiabatic approximation we have:

$$\partial_t \mathbf{u}' + \overline{\mathbf{U}} \cdot \nabla \mathbf{u}' + \mathbf{U}' \cdot \nabla \overline{\mathbf{u}} + NL_{\mathbf{u}} + f\mathbf{e}_z \times \mathbf{u}' = -\nabla_H p_*',$$

$$NL_{\mathbf{u}} = \mathbf{U}' \cdot \nabla \mathbf{u}' - \overline{\mathbf{U}' \cdot \nabla \mathbf{u}'}$$

(2.1a,b)

$$\partial_t b' + \overline{\mathbf{U}} \cdot \nabla b' + \mathbf{U}' \cdot \nabla \overline{b} + NL_b = 0, \qquad NL_b = \mathbf{U}' \cdot \nabla b' - \overline{\mathbf{U}' \cdot \nabla b'}$$

(2.2a,b)

where $\mathbf{U}(\mathbf{u}, w)$ is 3D velocity field, $\nabla$ is the 3D gradient operator, $\nabla_H$ is its horizontal component, $NL_{\mathbf{u},b}$ are the non-linear terms in corresponding equations, $p' = \rho_0 p_*'$ is eddy pressure, $\rho_0 = 10^3 \, kg \, m^{-3}$ is the reference density. Next, we multiply Eqs. (2.1a) and (2.2a) by $\mathbf{u}'$ and $b'/N^2$ accordingly and average over the sub-grid scale to derive the evolution equations for eddy kinetic and potential energy $K$, $W$:

$$\partial_t K + \overline{\mathbf{U}} \cdot \nabla K + \overline{\mathbf{U}' \cdot \nabla |\mathbf{u}'|^2}/2 = P_K,$$

$$P_K = P_K^e + P_K^L, \qquad P_K^e = -\overline{\mathbf{u}' \cdot \nabla_H p_*'}, \qquad P_K^L = -\text{trace}(\mathbf{R} \cdot \nabla \overline{\mathbf{u}}), \qquad K = \overline{|\mathbf{u}'|^2}/2$$

(2.3a,b,c,d,e)

$$\partial_t W + \overline{\mathbf{U}} \cdot \nabla W + \overline{\mathbf{U}' \cdot \nabla |b'|^2}/2N^2 = P_W,$$

$$P_W = -N^{-2} \overline{\mathbf{U}'b'} \cdot \nabla \overline{b}, \qquad W = \overline{|b'|^2}/2N^2$$

(2.4a,b,c))

where $\mathbf{R} = \overline{\mathbf{U}'\mathbf{U}'}$ is the eddy Reynolds stress, $P_{K,W}$ are production rates of EKE and EPE. In Eqs. (2.3a) and (2.4a) the second terms represent the advection of $K$ and $W$ by the mean flow while the third ones may be interpreted as their diffusion. Dubovikov and Canuto (2005) evaluated those terms and concluded that they are smaller than $P_{K,W}$. In this sense the eddy production is approximately a horizontally local process. Thus, below we neglect the advection/diffusion terms in Eqs. (2.3a) and (2.4a), i.e. consider the equations in the local approximation which allows us to parameterize eddy correlation functions in terms of large scale fields at the same locations. Next, the first term in (2.3b) is due the exchange EPE $\leftrightarrow$ EKE while the second one is the production of EKE by the mean flow. It is worth noticing that after averaging the direct production of EKE by MPE $-\overline{\mathbf{u}' \cdot \nabla_H \overline{p_*}}$ vanishes. This conclusion is in an agreement with the referred to above diagrams





of the energy exchange between mean flow and mesoscale eddies by Lorenz (Holton, 1992) and
Boning and Budich (1992) in which an exchange between EKE and MPE is absent.

Recall that Eqs. (2.1)-(2.4) correspond to the adiabatic approximation of the 2D eddy
turbulence which neglects the energy transfer to the 3D stratified turbulence (ST) occurring due to
the Kelvin-Helmholtz instability. Since ST is generated by the shear fluctuations, its production
$P_{ST}$ is at expense of EKE. With account for $P_{ST}$, energy equations (2.3), (2.4) in the horizontally
local approximation are modified as follows:

$$\partial_t K = P_K^{eff} \equiv P_K - P_{ST}, \qquad \partial_t W = P_W \qquad (2.5\text{a,b})$$

The ST kinetic energy further cascades to the Kolmogorov scale $\sim l_K$ where it is finally dissipated.
Thus, the present model (2.5) overcomes the framework of the adiabatic approximation.
Nevertheless, the relation $P_W = 0$ keeps its validity in the more accurate approximation (2.5).

Next, as we show in Appendix, in the ocean interior the second term of $P_K$ in (2.3b) which
represents the EKE production by mean flow, is small in comparison with the first one which is
due to the EPE $\rightleftarrows$ EKE conversion. Thus, we may adopt the approximation:

$$P_K \approx -\overline{\mathbf{u}' \cdot \nabla_H p'_*} \qquad (2.6)$$

As for the production term in (2.4), we present it in the following form:

$$P_W = \mathbf{F}_H \cdot \mathbf{s} - F_V \equiv \Sigma, \qquad \mathbf{s} = -N^{-2}\nabla_H \overline{b} \qquad (2.7\text{a,b})$$

where we use the notations $N^2 = \partial_z \overline{b}$ for the buoyancy frequency, $\mathbf{F}_H = \overline{\mathbf{u}'b'}$, $F_V = \overline{w'b'}$ for the
horizontal and vertical components of the buoyancy flux and $\mathbf{s}$ for the slope of the mean flow.
Transform $F_V$ in the hydrostatic approximation as follows:

$$F_V = \overline{w'b'} = \overline{w'\partial_z p_*} = \partial_z \overline{w'p_*} - \overline{p'_* w'_z} = \partial_z \overline{w'p'_*} + \overline{p'_* \nabla_H \cdot \mathbf{u}'} = \partial_z \overline{w'p_*} - \overline{\mathbf{u}' \cdot \nabla_H p'_*} + \nabla_H \cdot \overline{\mathbf{u}'p'_*} \quad (2.8)$$

and notice that the horizontal length scale of coarse resolution mean fields as well as of correlation
functions of eddy fields is of the order of $L \sim 10^3 km$ while that of mesoscale fields $\sim r_d$.
Therefore, the ratio of the last term of (2.8) to the penult one is $\sim r_d / L < 0.1$ and so it is negligible.
From Eqs. (2.6)-(2.8) we get the relations

$$P_W = -P_K + \mathbf{F}_H \cdot \mathbf{s} + R_W, \qquad R_W = -\partial_z \overline{w'p'_*} \qquad (2.9\text{a,b})$$

together with the relation for the production of the total eddy energy TEE=EKE+EPE

$$P_T = \mathbf{F}_H \cdot \mathbf{s} + R_W \qquad (2.10)$$



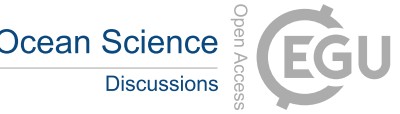

The interpretation of the terms in the right hand side of (2.9a) is obvious: the first term represents the transformation EKE $\rightleftarrows$ EPE, the second one determines the transition MPE $\rightarrow$ EPE due to the baroclinic instability of the mean flow, while the last term represents the redistribution of EPE between different layers. Indeed, the integral of that term over the ocean depth vanishes as it follows from the boundary condition which requires vanishing $w'$ at the ocean surface and bottom. Thus, the productions (2.6), (2.7) and (2.9) averaged over the ocean depth $H$ are:

$$\langle P_K \rangle \approx -\left\langle \overline{\mathbf{u}' \nabla_H p'} \right\rangle \approx \langle F_V \rangle, \qquad \langle P_W \rangle = -\langle F_V \rangle + \langle \mathbf{F}_H \cdot \mathbf{s} \rangle, \qquad \langle P_T \rangle = \langle \mathbf{F}_H \cdot \mathbf{s} \rangle, \qquad \langle R_W \rangle = 0$$

(2.11a-d)

where

$$\langle \bullet \rangle \equiv H^{-1} \int_{-H}^{0} \bullet \, dz \qquad\qquad (2.12)$$

## 3. Cascades of EKE and EPE

We recall that in idealized turbulent cascades the energy sources and sinks in wave-number space are separated by a rather extended inertial interval within which the energy production, transformation and dissipation are small. The sources and sinks are characterized by the rates of productions $P_{K,W}$ at the cascade beginnings and leakages $\varepsilon_{K,W}$ where the cascades terminate. Of course, in real flows the above conditions may be satisfied rather roughly. In the majority of 3D turbulent flows, the spectra of sources are concentrated at the largest scales of the energy spectrum while the sink is the viscous energy dissipation $\varepsilon_K$ occurring at Kolmogorov scales which are the smallest ones of the spectrum. The cascade sink $\varepsilon$ is not necessarily the viscous dissipation $\varepsilon$. For example, in 2D homogeneous turbulence, the sink of the inverse energy cascade occurs at the largest scales of turbulence. The energy cascades from the sources to the sinks are missed in the linear approximation since they are produced by the non-linear interactions in the Navier-Stokes equation (NSE). The dominating non-linear interactions are ones between modes with close wave numbers, i.e. the non-linear interactions are local in k-space. It means that the energy flows fluently from the source to sink like water in a pipe. On the basis of the similarity Kolmogorov found the celebrated energy spectrum $\sim k^{-5/3}$ for homogeneous isotropic turbulence. Analyzing the similarity in more details within the renormalization group approach, Dubovikov (1993), Canuto and Dubovikov (1996, 1997) developed Langevin-type equations for turbulent flows which have no adjustable parameters and yet with a satisfactory accuracy they yield spectra and turbulent statistics for a wide class of flows (Canuto and Dubovikov, 1996-1999).

As we notice in Introduction, in the case of 2D mesoscale turbulence there are the two energy cascades: the downscale cascade of EPE and the up-scale one of EKE. The EPE is generated by the baroclinic instability of the large scale flow, i.e. by mean potential energy (MPE) at scales $\sim r_d$ at which the Rossby number is small and, therefore, EPE cannot noticeably transform into EKE. As a result, the EPE cascades downscale until at $k \sim 1/\ell$ the spectral Rossby number $Ro(k)$ reaches unity that allows the conversion EPE $\rightarrow$ EKE. Thus, at scales $\sim \ell$ there occurs the leakage $\varepsilon_W(\ell)$ of EPE and the production $P_K(\ell)$ of EKE which in turn cascades up-scales. Further, as we notice in



Introduction, a sink of the EKE cascade at scales $\sim r_d$ may be either MKE or EPE. In Appendix we show that the later dominates that results in a pronounced maximum of EKE spectrum at $k \sim k_0 \sim 1/r_d$. Thus, at scales $\sim r_d$ the terminating downscale EKE cascade feeds EPE and reinforces the EPE cascade produced by the large scale baroclinic instability thus closing the mesoscale energy cycle. In summary, at the maximal scales of the mesoscale spectra $\sim r_d$ there are the production of EPE $P_W(r_d)$ and the leakage of EKE $\varepsilon_K(r_d)$ while at the minimal scales $\sim \ell$ there occur the production of EKE $P_K(\ell)$ and the leakage of EPE $\varepsilon_W(\ell)$. Since both $r_d$ and $\ell$ considerably exceed the Kolmogorov scales $\sim l_K$ at which the viscous dissipation $\varepsilon_K$ occurs, we conclude that the latter is negligible in the region of the mesoscale cascades $1/r_d < k < 1/\ell$ in comparison with the leakages $\varepsilon_K(r_d)$ and $\varepsilon_W(\ell)$ which feed the productions $P_W(r_d)$ and $P_K(\ell)$ correspondingly. As for $\varepsilon_K$, it is produced by the stratified turbulence, ST, which is generated by the Kelvin-Helmholtz instability at scales $\sim \ell$ as we discuss below. Thus, besides of the EKE and EPE cascades, there also exists the third, the ST down scale one between the scales $\sim \ell$ and $l_K$ which ultimately terminates due to the viscous dissipation at scales $\sim l_K$.

Because of the absence of the viscous dissipation within the whole interval of the mesoscale spectra between the scales $\sim \ell$ and $\sim r_d$, we may treat $\varepsilon_{K,W}$ as negative productions, i.e.

$$\varepsilon_{K,W}(\text{sink}) \equiv -P_{K,W}(\text{sink}) > 0 \qquad (3.1)$$

Since the existence of the EKE and EPE cascades imply that the spectra of their productions and sinks are concentrated at the different ends of the cascades, Eqs.(2.9) are satisfied at scales $\sim \ell$ and $\sim r_d$ separately. Then from (2.9a) with account for (3.1) and the fact that the spectrum of $\mathbf{F}_H \cdot \mathbf{s}$ is concentrated at scales $\sim r_d$, we obtain:

$$\varepsilon_K(r_d) \equiv -P_K(r_d) = P_W(r_d) - \mathbf{F}_H \cdot \mathbf{s} - R_W(r_d) > 0,$$
$$\varepsilon_W(\ell) \equiv -P_W(\ell) = P_K(\ell) - R_W(\ell) > 0 \qquad (3.2a,b)$$

The transition of $K, W$ from the source to the sink is characterized by the cascading fluxes (CF) $\Pi_{K,W}$ in wave number space which are directed from the sources to the sinks. In stationary flows we have

$$\Pi_{K,W} = P_{K,W}(\text{source}) = \varepsilon_{K,W}(\text{sink}) \equiv -P_{K,W}(\text{sink}) > 0 \qquad (3.3)$$

Since in Eq.(2.5) $P_{ST}$ is contributed at scales $\sim \ell$ only, at the scales $\sim r_d$ from (3.2), (3.3) we obtain the following relations for the productions and CFs:

$$\Pi_K = \varepsilon_K(r_d) \equiv -P_K(r_d) > 0,$$
$$\Pi_W = P_W(r_d) = -P_K(r_d) + \mathbf{F}_H \cdot \mathbf{s} + R_W(r_d) > 0 \qquad (3.4a,b)$$

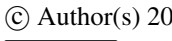



where $P_K$ is given in (2.6). The source of EKE cascade occurs at scales $\sim \ell$ and, therefore, from (2.5a) we have $P_K(\text{source}) = P_K(\ell) - P_{ST}$. Thus, from (3.3) we obtain:

$$\Pi_K = P_K(\ell) - P_{ST} > 0,$$
$$\Pi_W = -P_W(\ell) = \varepsilon_W(\ell) = P_K(\ell) + R_W(\ell) \tag{3.5a,b}$$

The outlined idealized scheme of the energy cascades and productions presented in (3.3)-(3.5), may be sketched by Fig.1 which demonstrates that at scales $\sim r_d$ where are the source of EPE and the sink of EKE, we have relations (3.4), while at scales $\sim \ell$ where are the source of EKE and the sink of EPE as well as the source of ST, we have relations (3.5).
Next, from (3.4) and (3.5) we get:

$$P_K = \mathbf{F}_H \cdot \mathbf{s} + R_W = P_{ST} = \Pi_{ST} = \varepsilon_{ST}, \qquad P_W = 0 \tag{3.6a,b}$$

where all the functions $A$ above are defined as follows:

$$A \equiv A(r_d) + A(\ell) \tag{3.7}$$

In the next section we list sequences of the realization of the outlined EPE $\rightleftarrows$ EKE cycle and its coherence with the generation of the ST cascade.

## 4. Coherence of the EPE, EKE and ST cascades

The coherence of the outlined EPE, EKE and ST cascades implies the following conditions:
1) Occurrence of the sink of the EKE cascade at scales $\sim r_d$ implies a negative production of EKE at those scales:

$$P_K(r_d) < 0 \tag{4.1}$$

2) At scales $\sim r_d$ the mesoscale Rossby number should be small to prevent the transformation EPE $\rightarrow$ EKE,

$$Ro \equiv K^{1/2} / (fr_d) \ll 1 \tag{4.2}$$

3) Since spectral integrated EPE does convert into EKE, we expect that at higher $k$ (smaller scales) the spectral Rossby number increases and at some wave numbers $\sim 1/\ell$ it reaches unity

$$Ro(1/\ell) \sim 1 \tag{4.3}$$

where



$$Ro(k) = kU(k)/f, \qquad \frac{1}{2}U^2(k) = \int_{k}^{k_{max}} E(k')dk' \qquad (4.4a,b)$$

where $k_{max}$ is the largest wave number of the mesoscale energy spectrum $E(k)$. At scales satisfying (4.3) the transformation process $EKE \rightarrow EPE$ changes its direction and EPE transforms into EKE.

4) To generate the ST cascade at the same scales $\sim \ell$, the spectral Richardson number should reach unity

$$Ri(1/\ell) \sim 1 \qquad (4.5)$$

The mesoscale spectral Richardson number relates to the spectrum of the mesoscale shear $\overline{S^2} \equiv \overline{\left| \partial_z \mathbf{u}' \right|^2}$ as follows:

$$Ri(k) = N^2 \left( \int_{k_{min}}^{k} \overline{S^2}(k)dk \right)^{-1} \qquad (4.6)$$

Under condition (4.5) the eddy turbulence creates the Kelvin-Helmholtz instability which generates ST which finally dissipates the eddy energy.

In the subsequent sections we show that within D3, CD5,6 models conditions (4.1), (4.2) are satisfied at scales $\sim r_d$ while at scales $\sim \ell$ determined by condition (4.3), condition (4.5) is satisfied as well.

## 5. Transformation EKE→EPE at scales $\sim r_d$

To analyze condition (4.1), we present $P_K$ (2.6) as follows:

$$P_K \approx f \overline{\mathbf{e}_z \times \mathbf{u}_g \cdot \mathbf{u}'} = f \overline{\mathbf{e}_z \times \mathbf{u}_g \cdot \mathbf{u}_a} = -f \overline{\mathbf{u}_a \times \mathbf{u}_g} \cdot \mathbf{e}_z,$$
$$\mathbf{u}_g = f^{-1} \mathbf{e}_z \times \nabla_H p'_*, \qquad \mathbf{u}_a = \mathbf{u}' - \mathbf{u}_g \qquad (5.1a,b,c)$$

where $\mathbf{u}_g (\mathbf{u}_a)$ are the geostrophic and a-geostrophic components of the eddy velocity. Consider the approximation of a stationary mean flow and infinite large scales in which limit mean fields are independent on horizontal coordinates and time while the mesoscale turbulence is stationary and horizontally homogeneous. Then in Fourier space relations (5.1) have the form:

$$P_K(\mathbf{k}, z)\delta(\mathbf{k} + \mathbf{k}') = -f \operatorname{Re}\left\{ \overline{\mathbf{u}_a(\mathbf{k}', z) \times \mathbf{u}_g(\mathbf{k}, z)} \right\} \cdot \mathbf{e}_z,$$

$$(5.2a,b,c)$$

$$\mathbf{u}_a(-\mathbf{k}) = \mathbf{u}_a^*(\mathbf{k}), \qquad P_K(z) = \int d^2\mathbf{k} P_K(\mathbf{k}, z)$$



In these formulae, following Killworth (1997, 2005) and D3, CD5, we use the same symbols $P_K$ and $\mathbf{u}'$ for the designation of the production and the mesoscale velocity fields in both physical and Fourier spaces. The analogous kind of notations are adopted below in all mesoscale equations and mesoscale fields and functions. The difference is that in Fourier space the independent variables are the wave vector $\mathbf{k}$ (or $k \equiv |\mathbf{k}|$) and the frequency $\omega$. Since in physical space $\mathrm{curl}\mathbf{u}_a = 0$ and $\mathrm{div}\mathbf{u}_g = 0$, in Fourier space in the approximation of axi-symmetric eddies we have the relations:

$$\mathbf{u}_a(\mathbf{k}) = \mathbf{n}u_a(k), \qquad \mathbf{u}_g(\mathbf{k}) = \mathbf{n} \times \mathbf{e}_z u_g(k), \qquad \mathbf{n} = \mathbf{k}/k \qquad \text{(5.3a,b,c)}$$

Equations of D3/CD5 model differ from those of the widely applied linear approximation by the presence of non-linear terms. In particular, in Eqs. (2.1a,b) for the mesoscale velocity in Fourier space in the vicinity of the maximum of the energy spectrum, i.e. at $k \sim 1/r_d$, the non-linear term equals

$$NL_{\mathbf{u}}(\mathbf{k}) = -\nu\mathbf{u}', \qquad \nu^{-1} \approx \mathrm{K}^{-1/2}\mathrm{r}_d \qquad \text{(5.4a,b)}$$

where the minus sign is due to the inverse direction of EKE cascade. After Fourier transformation of (2.1a) with respect to the horizontal coordinates and time, with use of (5.3), (5.4) and the dispersion relation $\omega = \mathbf{k} \cdot \mathbf{u}_d$ we obtain [see details of the derivation in D3, Eqs. (24b-d), and CD5, Eq.(10a), (15a)]:

$$-f u_a(k) = [\nu - i\mathbf{k} \cdot (\bar{\mathbf{u}} - \mathbf{u}_d + k^{-2}\mathbf{e}_z \times \boldsymbol{\beta})]u_g(k) \qquad \text{(5.5)}$$

where $\boldsymbol{\beta} = \nabla f$ and $\mathbf{u}_d$ is the eddy drift velocity whose expression in terms of large scale fields in a pure adiabatic ocean is given in Eq.(25m) of D3 and Eq.(4f) of CD6. Substituting (5.4) and (5.5) into (5.1a), we notice that the imaginary term of (5.5) does not contribute to the real part of $P_K(\mathbf{k}, z)$ (5.2a). Since at scales $\sim r_d$ the ageostrophic component of the mesoscale velocity is small, from (5.2), (5.3) and (5.5) we deduce the following relation for the spectrum of the EKE production at $k \sim 1/r_d$:

$$P_K(k) = -2\nu E(k), \qquad k \sim 1/r_d \qquad \text{(5.6a,b)}$$

Integrate this spectrum around $k \sim 1/r_d$ in the interval $\Delta k \sim 1/r_d$ in which the mesoscale energy spectrum is concentrated. With account for (5.4b) we obtain the EKE production at scales $\sim r_d$

$$P_K(r_d) = -2r_d^{-1}K^{3/2} < 0 \qquad \text{(5.7)}$$

which satisfies condition (4.1) and means that at the scales $\sim r_d$ EKE transforms into EPE. To get a pronounced maximum of $E(k)$ at those scales, the absolute value of $P_K(r_d)$ needs to exceed the





dissipation of mesoscale energy which practically coincides with the dissipation of ST in analogy with a weakly attenuating pendulum in which the transformation $PE \rightleftarrows KE$ exceeds the dissipation. To check the analogous condition for mesoscales, we adopt $r_d \approx 2\cdot10^4 m$, $K \approx 10^{-2}$ $m^2 s^{-2}$ and the ocean depth $H \sim 1 km$. Then from (5.7) we obtain the column production

$$H\left|\left\langle P_K(r_d)\right\rangle\right| \sim 10^{-4} m^3 s^{-3} \qquad (5.8)$$

Compare this result with the column production of TEE $P_T H$ using (2.11c) and the Gent-McWilliamc (1990, 1995, GM) large scale horizontal buoyancy flux:

$$\mathbf{F}_H^{GM} = -\kappa_M \nabla_H \bar{b} \qquad (5.9)$$

where $\kappa_M \sim 10^3 m^2 s^{-1}$ is the mesoscale diffusivity. We obtain

$$HP_T \sim \kappa_M N^2 \left|\mathbf{s}\right|^2 H \sim 10^{-5} m^3 s^{-3} \qquad (5.10)$$

From (5.8), (5.10) and (3.6a) we obtain the condition

$$\left\langle \left|P_K(r_d)\right|\right\rangle \gg \left\langle P_T\right\rangle = \left\langle P_{ST}\right\rangle = \left\langle \varepsilon_{ST}\right\rangle \qquad (5.11)$$

which ensures the realization of the pronounced maximum of EKE spectrum.

## 6. The transformation EPE→EKE at scales $\sim \ell$

As we noticed in Introduction, although at scales less than $\sim r_d$ we have no such a quantitative models as D3 and CD5, we are able to perform a semi-quantitative analysis in that region assuming that the slope of the energy spectrum deduced from the data of fine resolution simulations (Zhong and Bracco, 2013) and observations (Callies and Ferrari, 2013) at scales ~ few kilometers

$$E(k) \sim r_d K\left(r_d k\right)^{-m}, \quad m \sim 2 \qquad (6.1)$$

is maintained to scales of the order of tens meters. Then from (4.4) we obtain the velocity and the Rossby number at scales $\sim 1/k$:

$$U(k) \sim K^{1/2}\left(r_d k\right)^{-(m-1)/2}, \qquad Ro(k) \sim \left(f r_d\right)^{-1} K^{1/2}\left(r_d k\right)^{-(m-3)/2} \qquad (6.2a,b)$$

Thus, condition (4.3) is satisfied at scales $\ell$ for which

$$\left(\ell/r_d\right)^{3-m} \sim \left(r_d f\right)^{-2} K, \qquad m < 3 \qquad (6.3)$$





At those scales EPE finally releases into EKE and the down-scale EPE cascade terminates while the inverse EKE cascade starts up. Thus, we conclude that both EKE and EPE cascades occur in the wavenumber interval

$$1/r_d < k < 1/\ell \qquad (6.4)$$

If we adopt the idealized scheme of the conversion EPE→EKE which is concentrated at the scales $\sim \ell$ as sketched in Fig.1, the exponent in (6.1) is the Kolmogorov one $m = 5/3$. Then from (6.3) we have:

$$m = 5/3, \qquad \ell \sim f^{-3/2} r_d^{-1/2} K^{3/4} \sim 100 m \qquad (6.5)$$

where in the numerical evaluation we choose the typical values $f = 10^{-4} s^{-1}$, $r_d \sim 2 \cdot 10^4 m$, $K \sim 10^{-2} m^2 s^{-2}$. If the region of the conversion is somewhat spread above the scales $\sim \ell$, then $m > 5/3$. In particular, adopting m=2, from (6.3) we have

$$m = 2, \qquad \ell \sim r_d^{-1} f^{-2} K \sim 50 m \qquad (6.6)$$

## 7. Dissipation of TEE at $k \sim 1/\ell$

Next, we show that within the present model condition (4.5) also is satisfied, i.e. at the scales $\sim \ell$ the vertical eddy shear fluctuations $\overline{S^2} \equiv \overline{|\partial_z \mathbf{u}'|^2}$ create the Kelvin-Helmholtz instability which generates the stratified turbulence (ST) which finally dissipates eddy energy. To evaluate $\overline{S^2}$, we need its spectrum $\overline{S^2}(k)$. Since in the whole range (6.4) $Ro(k)$ does not exceed unity considerably, for evaluating the eddy shear we may use the geostrophic relation. Thus, in wavenumber space we obtain the evaluation:

$$\overline{S^2}(k) \sim f^{-2} k^2 \overline{b'^2}(k) \qquad (7.1)$$

The buoyancy variance spectrum is related to EPE spectrum $W(k)$ as follows

$$\overline{b'^2}(k) = 2N^2 W(k) \qquad (7.2)$$

Adopt

$$W(k) \sim E(k) \qquad (7.3)$$





that is true at scales $\sim r_d$. Substitute (7.3) , (7.2) and (6.1) into (4.6) and use the fact that the integral in (4.6) is contributed mostly by the region in the vicinity of the upper limit. With account for (6.2b) we obtain:

$$Ri(k) \sim f^2 k^{m-3} r_d^{m-1} K^{-1} \sim \left[Ro(k)\right]^{-1/2} \tag{7.4}$$

Substituting here relation (6.1), we conclude that $Ri(1/\ell)$ satisfies condition (4.5), and, therefore, the vertical eddy shear fluctuations create the Kelvin-Helmholtz instability which generates ST which finally dissipates the turbulence energy supplied by EKE. So at scale $\ell$ (6.1) not only the EPE cascade terminates and EPE releases into EKE generating the inverse EKE cascade, but in addition, at the same scales the eddy shear fluctuations generate ST which dissipates the eddy energy. Both evaluations of $\ell$ (6.5) and (6.6) are consistent with the observed horizontal scale of ST~50m (Muller et al., 2002). The column energy dissipation of ST $\varepsilon_{ST}$ approximately coincides with column production of TEE $P_T$. This conclusion is in agreement with result (3.6a).

## 8. The vertical scale of ST

The vertical scale of the ST coincides with the vertical displacement of isopycnals at horizontal scales $\sim \ell$ . From (7.1), (4.5) we deduce

$$\overline{z'^2}(1/\ell) \sim N^{-4}\overline{b'^2}(1/\ell) \sim N^{-4}f^2\overline{S^2}(1/\ell)\ell^2 \sim N^{-2}f^2\ell^2 \tag{8.1}$$

which together with (6.5), (6.6) is consistent with the observed vertical scale of ST

$$\left[\overline{z'^2}(1/\ell)\right]^{1/2} \sim 10m \tag{8.2}$$

 (see Muller et al., 2002).

## 9. Summary

Even though it is recognized that the generation of mesoscale eddies is the dominant sink of the large scales energy, it is only a path toward dissipation but not dissipation itself (Wunsch and Ferrari, 2004). The commonly recognized conundrum is "how is the dynamical transition made from a 2-D cascade at large scales-with its strong inhibition of down-scale energy flux-to the more isotropic, 3D-like, down-scale cascades at small scales" where the energy is dissipated (Muller et al., 2002; Mc Williams, 2003). In the present essay we have sketched the pathway to the dissipation in the framework of D3/CD5 mesoscale model within which we have discussed also other related problems of generation of the inverse energy cascade and its termination. As we sketch the process in Fig.1, the large scale baroclinic instability feeds EPE at scales $\sim r_d$. Since at those scales the mesoscale Rossby number is small, EPE cannot release into EKE and so it cascades to smaller



scales. Even more, D3/CD5 predicts the conversion EKE→EPE at the scales $\sim r_d$ that provides an additional EPE feeding and amplifies the downscale EPE cascade generated by the baroclynic instability. However, at smaller scales the spectral Rossby number (6.2) increases. Finally, at scales $\sim \ell \sim 100$ m (6.5) the spectral Rossby number reaches unity and so EPE converts into EKE which, in turn, cascades upscale and at scales $\sim r_d$ converts back into EPE closing the conversion cycle EPE $\rightleftarrows$ EKE. In addition, at scales $\sim \ell$ the fluctuations of the vertical shear create the Kelvin-Helmholtz instability which generates ST dissipating the eddy energy. We have evaluated $\ell$ under the assumption that the slope of the energy spectrum (6.1) deduced from the data of fine resolution simulations (Zhong and Bracco, 2013) and observations (Callies and Ferrari, 2013) until to scales $\sim$ few kilometers, is maintained at scale of the order of tens meters. In the steady state the eddy energy dissipation $\varepsilon$ integrated over the ocean depth balances the column production of TEE by large scale baroclinic instability $P_T$ (2.11c). At the non-balanced stage of flow development when $\int \varepsilon \mathrm{dz} < P_T$, eddy fields increase. Thus, if $\varepsilon$ were negligible, the eddy field would unlimitedly grow and the maximum of the energy spectrum would become more and more prominent until $\varepsilon$ will balances $\Pi_T$. Let us stress that the suggested pathway of the ocean energy from its generation to dissipation is not the only one in Word Ocean. In Introduction we have referred to the main mechanisms considered thus far all of which are quite different than the one presented above. However, in our opinion, the latter dominates. Still, we stress that the suggested cartoon is rather rough and idealized. In particular, within the range of perfect cascades, energy production and losses (i.e. the conversion EPE $\rightleftarrows$ EKE) would be absent that would result in Kolmogorov EKE spectrum (6.1) with $m = 5/3$. However, in reality in the whole interval $(1/r_d - \ell)$ some conversion occurs that results in a correction to the Kolmogorov $m = 5/3$.

## Aknowledgements

I thank Professor V.M. Canuto for fruitful discussions of the considered problems and Dr. M. Alexandrov for a valuable help in preparation of the manuscript. This work was supported by NASA High-End Computing (HEC) Program through the NASA Center for Climate Simulations (NCCS) at the Goddard Space Flight Center.

## Appendix . Contribution of mean flow to production of EKE

Consider EKE production by the horizontal (isopycnal) mean shear. It is more convenient to consider the problem in isopycnal coordinates where in the adiabatic approximation the mesoscale Reynolds stress and the mean shear are 2D tensors:

$$R_{ij} = \overline{u_i' u_j'} - \delta_{ij} K, \qquad S_{ij} = \left( \partial_j u_i - \partial_i u_j \right)/2 \qquad (A.1)$$

The EKE production by the horizontal mean shear equals:





$$P_K^{hor} = -R_{ij}S_{ij} \qquad (A.2)$$

Because of the approximate axi-symmetry of mesoscale eddies, we have $\left|R_{ij}\right| < K$ while $\left|S_{ij}\right| \sim \left|\bar{\mathbf{u}}\right| / L$ where $L \sim 10^6 m$ is the large scale. Thus we may evaluate (A.2) as follows:

$$P_K^{hor} < \left(\left|\bar{\mathbf{u}}\right| / L\right) K \qquad (A.3)$$

Substituting $\left|\bar{\mathbf{u}}\right| \sim 0.03 \text{ms}^{-1}$, $K \sim 10^{-2} m^2 s^{-2}$, we evaluate the column production

$$H\left\langle P_K^{hor} \right\rangle < 10^{-7} m^3 s^{-3} \qquad (A.4)$$

which in two orders of magnitude is less than the column production of TEE (5.10) generated by the baroclinic instability.

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

## Caption to the figure

**Fig.1. S**ketch of an idealized pathway of the ocean large scale energy to its dissipation at Kolmogorov scales via mesoscale cascades and their leakage into the stratified turbulence generated by the Kelvin-Helmholtz instability of mesoscale cascades. Solid arrows show local (in wavenumber space) energy productions while dashed ones show the cascading fluxes $\Pi_K$, $\Pi_W$ and $\Pi_{ST}$ whose relations to the local productions and dissipations at the ends of the cascades are given in Eqs. (3.4)-(3.6).





**Fig. 1**