# Peer review of "Mesoscale cascades and the “conundrum” of energy transfer from large to dissipation scales in an adiabatic ocean"

_Ocean Science, 2017_

## Referee Comment (RC1) · Anonymous Referee #1 · 19 Jul 2017

I must disclose that I have not properly read the entire manuscript, which is very hard to follow. However, it appears that it is clear even just from the introduction that the entire argument builds on assumptions that are very much at odds with the general understanding of mesoscale ocean turbulence.

The introduction lists 7 "inputs" which are used to construct the argument. Multiple of these "input" assumptions appear to be strongly at odds with what I thought to be widely recognized.

For example, I would strongly disagree with assumption (3), that "intense release EPE-> EKE begins at scales where the spectral Rossby number Ro(k) which at large scale is

small, increases to unity." While the author argues that this is "commonly recognized", I believe that it is instead widely recognized that strong conversion from EPE->EKE occurs at the radius of deformation, which is generally much larger. (see e.g. the textbook by Geoff Vallis).

Maybe even more importantly, it is widely recognised (again see e.g. the textbook by Geoff Vallis) that total eddy energy is transferred to larger scales (both horizontally and vertically) in geostrophic turbulence (and mesoscale ocean turbulence is to a good approximation geostrophic). This appears to a priori undermine the entire argument that is attempted here.

Maybe I'm missing something important, but at the very least the author would have to clarify where his assumptions come from, why they appear to be glaringly at odds with well accepted theories, and why the reader should not be worried about this.

---

## Author Comment (AC1) · 2 Aug 2017

Reply to the comments of the Referee1 on "Mesoscale cascades and the "conundrum" of energy transfer from large to dissipation scales in an adiabatic ocean" M.S. Dubovikov First of all, I would like to thank the Referee for emphasizing the fact that the results of the manuscript under the discussion "are very much at odds with the general understanding of mesoscale turbulence". However, namely "the general understanding" leads to one of the most enigmatic conundrums of ocean general circulation which is "how does the energy of the general circulation cascade from the large climate scales, where most of it is generated, to the small scales, where all of it

is dissipated? In particular, how is the dynamical transition made from an anisotropic, 2D-like, geostrophic cascade at large scales-with its strong inhibition of down-scale energy flux-to 3D-like, down-scale cascades at small scales." (Muller et al., 2002). Specifically, the Referee states that: (1). "it is widely recognized that strong conversion EPEïĆőEKE occurs at the radius of deformation (see e.g. the text book by Geoff Vallis)". Indeed, in the chapter 6.8 titled "The energetics of linear baroclinic instability" Vallis studies the problem of the baroclinic instability and concluded in the end of the chapter that "baroclinic instability converts potential energy into kinetic energy." This conclusion was drawn on the basis of the linear analysis within which the energy exchange between different Fourier modes is absent at all, as well as the energy cascades. Meanwhile, those phenomena and the non-linear (NL) interactions are crucial for the mesoscale dynamics and observational effects, as Dubovikov (2003, D3) and Canuto and Dubovikov (2005, CD5) showed theoretically. An analogous conclusion was drawn by Chelton et al. (2011) from the analysis of observational data: "essentially all of the observed mesoscales features are non-linear", "mesoscales do not move with the mean velocity but with their own drift velocity" and the latter is "the most germane of all the non-linear metrics". In D3 and CD5 we derived the mesoscale drift velocity d u theoretically. In Fig.1 borrowed from Canuto et al. (2017a), we present the comparison of the predicted d u with observational data which were obtained later (Fu, 2009; Chelton and Schlax, 2013). In D3 and CD5 we parameterized the NL terms of the dynamical mesoscale equations on the basis of the general approach to modeling NL interactions in turbulent flows developed by the authors before (see the list of those articles in the manuscript under the discussion). The basis of the D3, CD5 mesoscale 2 parameterization is the generation of the inverse energy cascade in mesoscale turbulence whose existence is now commonly recognized (Ferrari and Wunsch, 2009; Bruggemann and Eden, 2015; Jansen et al., 2015) and confirmed by sea surface height data (Scott and Wang, 2005; Scott and Arbic, 2007). As Kraich-nan (1975) showed, that cascade generates the negative turbulent viscosity which drastically changes the mesoscale equations whose solution has no fitting parameters

and can be tested against data of observations and OGCMs numerical computations. Some validations of D3, CD5 are demonstrated below in Figs.1-3 borrowed from the submitted papers by Canuto et al. (2017a,b). Thus, we expect that the NL mesoscale dynamics radically modifies the transformation of EPE and EKE in comparison with the results of the linear analysis presented in the quoted above Vallis's text book. In particular, consider Eq.(5.7) of the manuscript under discussion which yields the EKE production ( ) K d P r by EPE at scales of the deformation radius d r : 1 3/2 ( ) 2 0 K d d P r r K ïĂ▪ ïĂ¡ ïĂ▪ ïĂij (a) where K is EKE. The mesoscale characteristics K and d r demonstrate the fact that ( ) K d P r is due to the cascades, i.e. due to the NL interaction. The negative sign in Eq.(a) means that at scales   d r EKE transforms into EPE. By contrast, at scales   given by Eq.(6.3) we have the conversion EPEïĆőEKE. The sign of the total EKE production PK (total) ïĂ¡ PK (rd ) ïĂń PK ( ) given in (5.10), is positive. Even without any mesoscale model it is clear that the negative sign of ( ) K d P r straightforwardly follows from the existence of the strong inverse energy cascade and the observational fact that the transfer of EKE to large scales is much less than the energy exchange between EKE and EPE. The latter follows from the oceanic analog of the observed atmospheric Lorenz (1960) energy cycle summarized by Holton (1992), Fig.10.13 adapted from Oort and Peixoto (1974). The same conclusion follows from the numerical simulations by Boning and Budich (1992, Figs. 8,9). The result (a) is odd with the discussed statement of the Referee cited in the beginning of (1). (2). The Referee states that "it is widely recognized that total eddy energy is transferred to larger scales". This is not correct. Exactly the opposite is true: the total eddy energy is fed mostly by the large scale available potential energy which is due to the baroclinic instability. Specifically, the production of EPE which ultimately converts into EKE and finally is dissipated, is mostly contributed by the transfer of available potential energy from large scales, the conclusion which 3 follows from, say, the Gent-McWilliams model as well as from D3 and CD5 ones. Thus, the large scale energy is transferred to the total eddy energy. (3). The Referee "strongly disagrees" with our input that "intense release EPEïĆőEKE begins at scales where the spectral

Rossby number Ro(k) which at large scales is small, increases to unity". Nevertheless, this conclusion follows straightforwardly from mesoscale equations (5e)-(6b) of D3 or Eqs.(4i) –(5b) of CD5 which account for NL terms. In fact, EKE is produced by the a-geostrophic component of the velocity a u . In Fourier space we have: * * ( ) Im ( ) ( ) K a P k ïĂ¡ ïĂ▪ ïČ ' ïČń p k k ïČŮu k ïČ ž ïČ ż (b) where 0 * p ïĂ¡ ïĄš p is the pressure, 3 3 0 ïĄš ïĂ¡10 kg / m is the reference density. In the case of a small Ro(k) from the referred above equations of D3 or CD5 to the main order of EKE using the manuscript notations we deduce: * ( ) ( ) ( ), ( ) a z g z g u k ïĂ¡ ïĂ▪Ro k e ïČ ťu k fe ïČ ťu k ïĂ¡ ïĂ▪ikp (c) where z e is the unit vertical vector, f is the Coriolis parameter. From Eqs.(b), (c) we get 2 1 2 * ( ) ( ) ( ) 0 K P k f Ro k p ïĂ▪ k ïĂ¡ ïĂ▪ k ïĂij (d) i.e. at small Ro(k) EKE transforms into EPE but not vice verca. It is worth recalling that this result is obtained with account for the negative turbulent viscosity in the referred mesoscale equations which, in turn, is due to the inverse energy cascade created by NL interactions which is absent in the linear approximation. In the opposite case of a large Ro(k) the effect of rotation is weak and the velocity equation yields the usual EPEïČ őEKE conversion. References Boning, C.W. and Budich, R.G., 1992 Eddy dynamics in a primitive equation model: sensitivity to horizontal rezolutionand friction. J. Phys. Oceanogr., 22, 361- 381. Bruggemann, N. and Eden, C., 2015 Routes to dissipation under different dynamical conditions. J. Phys. Oceanogr., 45, 2149-2168. 4 Canuto, V.M. and. Dubovikov, M.S., 2005 Modeling mesoscale eddies, Ocean Model., 8, 1-30, cited CD5. Canuto, V.M., M.S.Dubovikov, Y. Cheng, A.M. Howard, 2017a Parameterization of mixed layer and deep ocean mesoscales including non-linearity, , J. Phys. Oceanogr., under revision. Canuto V.M., M.S.Dubovikov, Y.Cheng and A.M.Howard, 2017b Mesoscale diffusivity: a location and depth dependent model. J. Phys. Oceanogr., to be submitted. Chelton, D.B., M.G.Schlax and R.M.Samelson, 2011 Global observations of non-linear mesoscale eddies, Progress in Oceanography, 91, 167-216 Dubovikov, M.S., 2003 Dynamical model of mesoscale eddies. Geophys. Astrophys Fluid Dyn., 7, 311-358. Ferrari, R. and Wunsch, C., 2009 Ocean circulation kinetic energy: reservoirs, sources, and sinks. Annu. Rev. Fluid. Mech., 41,

[Figure]

253-282. Fu, L.L., 2009 Patterns and velocity of propagation of the global ocean eddy variability, .J. Geophys .Res., 114, C11017, doi:10.1029/2009JC005349. Holton, J.R., 1992 An Introduction to Dynamic Meteorology. Academic Press, Inc. Jansen, M.F., Adcroft, A.J., Hallberg, R, Held, I.M, 2015 Parameterization of eddy fluxes based on a mesoscale energy budget. Ocean Model. 92, 28-41. Kraichnan, R.H., 1975 Statistical dynamics of two-dimensional flow. J. Fluid Mech., 67, 155-171. Lorenz, E., Energy and numerical weather prediction, Tellus, 1960, 12, 364-373. Muller, P., J. McWilliams, and Molemaker, 2002 Routes to Dissipation the Ocean: The 2D/3D Turbulence Conundrum. In H. Baurmert, J. Simpson, and J. Sundermann, editors, Marine Turbulence – Theories, Observations and Models. Results of the CARTUM Project. Cambridge Press. Phillips, H.E. and S.R.Rintoul, 2000 Eddy variability and energetics from direct measurements in the ACC south of Australia, J. Phys. Oceanogr, 30, 3050-3076 5 Scott, R.B. and Arbic, B.K., 2007 Spectral energy fluxes in geostrophic turbulence: implications for ocean energetics. J. Phys. Oceanogr., 37, 673-688. Scott, R.B. and Wang, F., 2005 Direct evidence of an oceanic inverse kinetic energy cascade from satellite altimetry. J. Phys. Oceanogr., 35, 1650-1666. Smith, K.S. and J.Marshall, 2009 Evidence for enhanced eddy mixing at middepth in the Southern Ocean, J. Phys. Oceanogr., 39, 50-69 WOCE Data Products Committee, 2002, WOCE Global Data, Version 3.0, WOCE International Project Office, WOCE Report No. 180/02, Southampton, UK. 6

Fig.1. Borrowed from Canuto et al. (2017a). Comparison of | d u | derived in D3 and CD5 with the data of Fu (2009) and Chelton and Schlax (2011) at 0 150 W and 0 110 W. The data are reproduced satisfactorily. In all the figures, the model results were obtained from an average of the last 3 years of a simulation with the GISS ER stand-alone OGCM which was run for 300 years. 7 Fig.2. Borrowed from Canuto et al. (2017b). Comparison of the z-profile of the EKE derived in d3 and CD5 in units of its surface value vs. WOCE data in different locations. The model results reproduce the data satisfactorily. 8 Fig.3 Borrowed from Canuto et al. (2017b). Comparison of mesoscale diffusivity (in m2s-1) computed within D3 and CD5 model vs. the measured data of Philips and Rintoul (2000, PR00) in the ACC (143E,

51S). The ïĄű ïĂ¡1 case is with the contribution of corrections of the higher order in the small parameter equal to the ratio (mean K/EKE). 9

Please also note the supplement to this comment:
https://www.ocean-sci-discuss.net/os-2017-23/os-2017-23-AC1-supplement.pdf

———————————————

[Figure]

[Figure]

**Fig. 1.** Borrowed from Canuto et al. (2017a). Comparison of | d u | derived in D3 and

[Figure]

**Fig. 2.** Borrowed from Canuto et al. (2017b). Comparison of the z-profile of the EKE derived in d3 and CD5 in units of its surface value vs. WOCE data in different locations. The model results reproduce the da

[Figure]

**Fig. 3.** Borrowed from Canuto et al. (2017b). Comparison of mesoscale diffusivity

---

## Author Comment (AC2) · 10 Aug 2017

Reply to the comments of the Referee#1

First of all, I would like to thank the Referee for emphasizing the fact that the results of the manuscript under the discussion "are very much at odds with the general understanding of mesoscale turbulence". However, namely "the general understanding" leads to one of the most enigmatic conundrums of ocean general circulation which is "how does the energy of the general circulation cascade from the large climate scales, where most of it is generated, to the small scales, where all of it is dissipated? In particular, how is the dynamical transition made from an anisotropic,

2D-like, geostrophic cascade at large scales-with its strong inhibition of down-scale energy flux-to 3D-like, down-scale cascades at small scales." (Muller et al., 2002. List of references see in the supplement). Specifically, the Referee states that: (1). "it is widely recognized that strong conversion EPE to EKE occurs at the radius of deformation (see e.g. the text book by Geoff Vallis)". Indeed, in the chapter 6.8 titled "The energetics of linear baroclinic instability" Vallis studies the problem of the baroclinic instability and concluded in the end of the chapter that "baroclinic instability converts potential energy into kinetic energy." This conclusion was drawn on the basis of the linear analysis within which the energy exchange between different Fourier modes is absent at all, as well as the energy cascades. Meanwhile, those phenomena and the non-linear (NL) interactions are crucial for the mesoscale dynamics and observational effects, as Dubovikov (2003, D3) and Canuto and Dubovikov (2005, CD5) showed theoretically. An analogous conclusion was drawn by Chelton et al. (2011) from the analysis of observational data: "essentially all of the observed mesoscales features are non-linear", "mesoscales do not move with the mean velocity but with their own drift velocity" and the latter is "the most germane of all the non-linear metrics". In D3 and CD5 we derived the mesoscale eddy drift velocity theoretically. In Fig.1 presented in the supplement we compare the predicted drift velocity with observational data which were obtained later (Fu, 2009; Chelton and Schlax, 2013). In D3 and CD5 we parameterized the NL terms of the dynamical mesoscale equations on the basis of the general approach to modeling NL interactions in turbulent flows developed by the authors before (see the list of those articles in the manuscript under the discussion). The basis of the D3, CD5 mesoscale parameterization is the generation of the inverse energy cascade in mesoscale turbulence whose existence is now commonly recognized (Ferrari and Wunsch, 2009; Bruggemann and Eden, 2015; Jansen et al., 2015) and confirmed by sea surface height data (Scott and Wang, 2005; Scott and Arbic, 2007). As Kraichnan (1975) showed, that cascade generates the negative turbulent viscosity which drastically changes the mesoscale equations whose solution has no fitting parameters and can be tested against data of observations and

OGCMs numerical computations. Some validations of D3, CD5 are demonstrated in Fgs.1-3 presented in the supplement. Thus, we expect that the NL mesoscale dynamics radically modifies the transformation of EPE and EKE in comparison with the results of the linear analysis presented in the quoted above Vallis's text book. In particular, consider Eq.(5.7) of the manuscript under discussion which yields the EKE production by EPE at scales of the deformation radius Rd (see Eq.(a) in the supplement). The negative sign in (5.7) and Eq.(a) which is due to the cascades, i.e. due to the NL interaction, means that at scales ∼Rd EKE transforms into EPE. By contrast, at scales l∼100m given by Eq.(6.3) in the manuscript under the discussion we have the conversion EPE to EKE. As a result, the sign of the total EKE production given in (5.10), is positive. Even without any mesoscale model it is clear that the negative sign of the EKE production at scales ∼Rd straightforwardly follows from the existence of the strong inverse energy cascade and the observational fact that the transfer of EKE to large scales is much less than the energy exchange between EKE and EPE. The latter follows from the oceanic analog of the observed atmospheric Lorenz (1960) energy cycle summarized by Holton (1992), Fig.10.13 adapted from Oort and Peixoto (1974). The same conclusion follows from the numerical simulations by Boning and Budich (1992, Figs. 8,9). The conclusion on the negative sign of the EKE production at scales ∼Rd is odd with the discussed statement of the Referee cited in the beginning of (1). (2). The Referee states that "it is widely recognized that total eddy energy is transferred to larger scales". This is not correct. Exactly the opposite is true: the total eddy energy is fed mostly by the large scale available potential energy which is due to the baroclinic instability. Specifically, the production of EPE which ultimately converts into EKE and finally is dissipated, is mostly contributed by the transfer of available potential energy from large scales, the conclusion which follows from, say, the Gent-McWilliams model as well as from D3 and CD5 ones. Thus, the large scale energy is transferred to the total eddy energy. (3). The Referee "strongly disagrees" with our input that "intense release EPE to EKE begins at scales where the spectral Rossby number Ro(k) which at large scales is small, increases to

unity". In the supplement on the basis of an analysis of NL interactions we prove the validity of that conclusion. Even without prove it is clear that in the case of a large Ro(k) the effect of rotation is weak and the velocity equation yields the usual EPE to EKE conversion.

Please also note the supplement to this comment:
https://www.ocean-sci-discuss.net/os-2017-23/os-2017-23-AC2-supplement.pdf

---

## Referee Comment (RC2) · Anonymous Referee #2 · 23 Aug 2017

The manuscript "Mesoscale cascades and the conundrum of energy transfer from large to dissipation scales in an adiabatic ocean" by M. Dubovikov tries the resolve the dilemma of missing processes for the dissipation of balanced flow. The manuscript is reasonably well written such that it is possible to follow, although the details of the key argument are as usual buried in many other papers by the author. The proposed solution to the dilemma appears to be simple: while kinetic energy cascades upscale, potential energy cascades downscales and takes up the upscale kinetic energy towards dissipation at small scales.

Key in the argumentation is that the conversion of EPE to EKE depends on the scale.

It is argued that when the spectral Rossby number (Ro) is small, there is no conversion from EPE to EKE (in fact it is shown to be directed from EKE to EPE). This is assumed to be the case at the Rossby radius where most of the eddy energy is produced by baroclinic instability. Instead, the EPE then cascades downscale until the spectral Ro increases such that conversion EPE to EKE can take place. Then the EKE cascades upscale until it is converted to EPE at large scales and small Ro again.

It is demonstrated that from the turbulence closure approach by Canuto, Dubovikov et al it follows that there is indeed conversion from EKE to EPE at scales like the Rossby radius. For the dominant conversion from EPE to EKE at smaller scales with large Ro no strong arguments are given as far as I can see. The directions of the energy cascades are apparently assumed but in agreement with common believe. I haven't followed the derivation of the key result Eq. 5.7 in detail, but it seems correct.

This is a nice idea of the meso-scale energy cycle. The inverse energy cascade would just be a closed loop within a larger scope with a cascade of total energy towards small scale. However, it must be wrong since it is against all believes, observational and in particular modelling results that energy conversion by baroclinic instability is directed from EKE to EKE at the scale of the Rossby radius. In fact in all studies I know of, it is directed from EPE to EKE at all scales. The only conclusion for me is that this prediction of the turbulence closure approach by Canuto, Dubovikov et al given by Eq. 5.7 is wrong. That does not mean that all predictions and the whole closure is wrong, but it shows that one has to be careful with implications of simple closures like that.

---

## Author Comment (AC3) · 14 Sep 2017

Reply to the comments of the Referee#2 (R2)

First of all, I would like to thank R2 for his extensive work in examining and critically reviewing the manuscript. R2 clearly and concisely interpreted the basic features of the presented mesoscale energy cycle: "it is demonstrated that from the turbulence closure approach by Canuto, Dubovikov et al it follows that there is indeed conversion from EKE to EPE at scales like the Rossby radius" and that "the directions of the energy cascades are apparently assumed but in agreement with common believe." R2 recognizes some progress by the presented scheme: "This is a nice idea of the meso-scale energy

cycle. The inverse energy cascade would just be a closed loop within a larger scope with a cascade of total energy towards small scale." The only objection of R2 is as follows: "However, it must be wrong since it is against all believes that energy conversion by baroclinic instability is directed from EPE to EKE at the scale of the Rossby radius. In fact in all studies I know of, it is directed from EPE to EKE at all scales." Indeed, in the chapter 6.8 titled "The energetics of linear baroclinic instability" of the well-known book by Vallis "Atmospheric and Ocean Fluid Dynamics" the author studies the problem of the baroclinic instability and concluded in the end of the chapter that "baroclinic instability converts potential energy into kinetic energy." However, this conclusion was drawn on the basis of the linear analysis within which the energy exchange between different Fourier modes is absent altogether, as well as the energy cascades. As for numerical models, in all of them the energy transformation was studied for one-point quadratic functions like EKE and EPE rather than for two-points correlation functions and their spectra. The production of EKE spectrum is contributed by the work of both the linear force and that of the non-linear (NL) one, which is referred to in turbulence studies as "transfer". Since the integral over the spectrum of the transfer exactly vanishes, the full (spectrum integrated) EKE production is determined by the linear force only. Therefore, the energy transition "is directed from EPE to EKE", as R2 correctly states. Nevertheless, in the case of strong turbulence in certain wavenumber ranges the modulus of the transfer considerably exceeds that of the work of the linear force and, thus, the production of the EKE spectrum in those ranges is determined mostly by the transfer. This means that in the most part of the range where the works of the linear and NL forces have opposite signs, EKE transforms to EPE. This conclusion does not depend on a concrete closure and refutes R2's statement that the energy conversion "is directed from EPE to EKE at all scales".

In the real Ocean, as Dubovikov (2003, D3) and Canuto and Dubovikov (2005, CD5) showed theoretically, the mesoscale turbulence is strong and the NL interactions determine the mesoscale dynamics. The analogous conclusion was drawn by Chelton et al. (2011) from the analysis of observational data: "essentially all of the observed

mesoscales features are non-linear". Thus, we expect that the NL mesoscale dynamics radically modifies the transformation of EPE and EKE in comparison with the linear analysis and there is a scale range in the mesoscale spectrum where EKE transforms to EPE against the common preconception. In the supplement we show that within the closure developed in D3 and CD5, EKE transforms to EPE namely at the scales of the deformation radius. Even though R2 criticizes this result, I thank him for the remark that his criticism "does not mean that all predictions and the whole closure is wrong". To confirm this statement, in the supplement I show also some other appreciable results of the closure which are compared favorably with observations. The closure is notting more than the old-time mixing length model reformulated in wave-number terms. The basis of it is the generation of the inverse energy cascade in mesoscale turbulence whose existence is now commonly recognized (Muller et al., 2005; Ferrari and Wunsch, 2009; Bruggemann and Eden, 2015; Jansen et al., 2015) and confirmed by sea surface height data (Scott and Wang, 2005; Scott and Arbic, 2007). As Kraichnan (1975) showed and as it was confirmed in numerous direct numerical simulations (DNS) and large eddy simulations (LES), such cascades yield the negative turbulent viscosity which drastically changes outputs of the mesoscale dynamics which can be tested against data of observations and OGCMs numerical computations.

References Bruggemann, N. and Eden, C., Routes to dissipation under different dynamical conditions. J. Phys. Oceanogr., 2015, 45, 2149-2168. Canuto, V.M. and. Dubovikov, M.S., Modeling mesoscale eddies, Ocean Model., 2005, 8, 1-30, CD5. Chelton, D.B., M.G.Schlax and R.M.Samelson, 2011, Global observations of non-linear mesoscale eddies, Progress in Oceanography, 91, 167-216 Dubovikov, M.S., Dynamical model of mesoscale eddies. Geophys. Astrophys Fluid Dyn., 2003,.7, 311-358, D3. Ferrari, R. and Wunsch, C., Ocean circulation kinetic energy: reservoirs, sources, and sinks. Annu. Rev. Fluid. Mech., 2009, 41, 253-282. Jansen, M.F., Adcroft, A.J., Hallberg, R, Held, I.M, Parameterization of eddy fluxes based on a mesoscale energy budget. Ocean Model., 2015, 92, 28-41. Kraichnan, R.H., Statistical dynamics of two-dimensional flow. J. Fluid Mech., 1975, 67, 155-171.

Muller, P., J. McWilliams, and Molemaker, Routes to Dissipation the Ocean: The 2D/3D Turbulence Conundrum, 2005. In Marine Turbulence – Theories, Observations and Models. Edited by H. Baurmert, J. Simpson, and J. Sundermann, pp.397-405, Cambridge University Press. Scott, R.B. and Arbic, B.K., Spectral energy fluxes in geostrophic turbulence: implications for ocean energetics. J. Phys. Oceanogr., 2007, 37, 673-688. Scott, R.B. and Wang, F., Direct evidence of an oceanic inverse kinetic energy cascade from satellite altimetry. J. Phys. Oceanogr., 2005, 35, 1650-1666. G.K. Vallis, "Atmospheric and Ocean Fluid Dynamics", 2012, Cambridge University press.

Please also note the supplement to this comment:
https://www.ocean-sci-discuss.net/os-2017-23/os-2017-23-AC3-supplement.pdf

**Supplement:**

**Work of the non-linear force within the D3, CD5 model**

Before discussing the transfer within D3, CD5 model we present some appreciable numerical results within the model and their comparison with observational data. As it was shown theoretically in D3 and CD5, the non-linear (NL) interactions are crucial for the mesoscale dynamics and observational effects. An analogous conclusion was drawn by Chelton et al. (2011) from the analysis of observational data: "*essentially all of the observed mesoscales features are non-linear*" including mesoscale eddies which "*do not move with the mean velocity but with their own drift velocity*". This conclusion agrees with the observation of Richardson (1993) that mesoscale eddies are "*water-mass anomalies that have nearly circular flow around their centers and that survive for many rotations and may move through the background water at speeds and directions inconsistent with background flow*" The C11 observational result for the drift velocity is in agreement with the earlier theoretical predictions of D3 and CD5 (see, Canuto et al., 2017a). In Canuto et al. (2017b) are presented also other theoretical results of D3 and CD5 which are determined by NL interactions and compared favorably with observational data. In Fig.1 borrowed from Canuto et al. (2017a), we present the comparison of the predicted drift velocity with observational data which were obtained later (Fu, 2009; Chelton and Schlax, 2013). In D3 and CD5 we parameterized the NL terms of the dynamical mesoscale equations on the basis of the general approach to modeling NL interactions in turbulent flows developed by the authors before (see the list of those articles in the manuscript under the discussion). Some validations of D3, CD5 are demonstrated below in Fgs.1-3 borrowed from the submitted papers by Canuto et al. (2017a,b). The NL mesoscale dynamics radically modifies the transformation of EPE and EKE in comparison with the results of the linear analysis presented in the quoted above Vallis's text book. In particular, consider Eq.(5.7) of the manuscript under discussion which yields the EKE production $P_K(r_d)$ by EPE at scales of the deformation radius $r_d$ :

$$P_K(r_d) = -2r_d^{-1}K^{3/2} < 0 \qquad\qquad (a)$$

where $K$ is EKE. The mesoscale characteristics $K$ and $r_d$ demonstrate the fact that $P_K(r_d)$ is due to the cascades, i.e. due to the NL interaction. The negative sign in Eq.(a) means that at scales $\sim r_d$ EKE transforms into EPE. In fact, EKE is produced by the a-geostrophic component of the velocity $\mathbf{u}_a$. In Fourier space we have:

$$P_K(\mathbf{k}) = -\mathrm{Im}\left[ p_*(\mathbf{k})\mathbf{k}\cdot\mathbf{u}_a^*(\mathbf{k}) \right] \qquad\qquad \text{(b)}$$

where $p = \rho_0 p_*$ is the pressure, $\rho_0 = 10^3\,kg/m^3$ is the reference density. In the case of a small $Ro(k)$ from the referred above equations of D3 or CD5 to the main order of EKE using the manuscript notations we deduce:

$$\mathbf{u}_a(\mathbf{k}) = -Ro(k)\mathbf{e}_z \times \mathbf{u}_g(\mathbf{k}), \qquad f\mathbf{e}_z \times \mathbf{u}_g(\mathbf{k}) = -i\mathbf{k}p_* \qquad\qquad \text{(c)}$$

where $\mathbf{e}_z$ is the unit vertical vector, $f$ is the Coriolis parameter. From Eqs.(b), (c) we get

$$P_K(\mathbf{k}) = -k^2 f^{-1} Ro(k)\left| p_*(\mathbf{k}) \right|^2 < 0 \qquad\qquad \text{(d)}$$

i.e. at small $Ro(k)$ EKE transforms into EPE but not vice verca. It is worth recalling that this result is obtained with account for the negative turbulent viscosity in the referred mesoscale equations which, in turn, is due to the inverse energy cascade created by NL interactions which is absent in the linear approximation. In the opposite case of a large $Ro(k)$ the effect of rotation is weak and the velocity equation yields the usual EPE→EKE conversion.

**References**

Canuto, V.M. and. Dubovikov, M.S., 2005 Modeling mesoscale eddies, *Ocean Model.,* **8,** 1-30, cited CD5.

Canuto, V.M., Y. Cheng, M.S.Dubovikov, A.M. Howard, 2017a Parameterization of mixed layer and deep ocean mesoscales including non-linearity, , *J. Phys. Oceanogr.*, submitted after revision.

Canuto, V.M., Y. Cheng, M.S.Dubovikov, A.M. Howard, 2017b Mesoscale diffusivity: a location and depth dependent model. *J. Phys. Oceanogr.*, to be submitted.

Chelton, D.B., M.G.Schlax and R.M.Samelson, 2011 Global observations of non-linear mesoscale eddies, *Progress in Oceanography*, **91**, 167-216

Dubovikov, M.S., 2003 Dynamical model of mesoscale eddies. *Geophys. Astrophys Fluid Dyn.,* **7**, 311-358.

Fu, L.L., 2009, Patterns and velocity of propagation of the global ocean eddy variability. *J.Geophys.Res.*, **114,** C11017, doi:10.1029/2009JC005349.

Kraichnan, R.H., 1975 Statistical dynamics of two-dimensional flow. *J. Fluid Mech.*, **67**, 155-171.

Muller, P., J. McWilliams, and Molemaker, 2005 Routes to Dissipation the Ocean: The 2D/3D Turbulence Conundrum. In H. Baurmert, J. Simpson, and J. Sundermann, editors, *Marine Turbulence – Theories, Observations and Models. Results of the CARTUM Project*. Cambridge Press.

Phillips, H.E. and S.R.Rintoul, 2000 Eddy variability and energetics from direct measurements in the ACC south of Australia, *J. Phys. Oceanogr,* **30,** 3050-3076

Richardson, P.L., 1993 Tracking ocean eddies, *American Scientist,* **81**, 261-271.

Scott, R.B. and Arbic, B.K., 2007 Spectral energy fluxes in geostrophic turbulence: implications for ocean energetics. *J. Phys. Oceanogr.*, **37**, 673-688.

Scott, R.B. and Wang, F., 2005 Direct evidence of an oceanic inverse kinetic energy cascade from satellite altimetry. *J. Phys. Oceanogr.*, **35**, 1650-1666.

Smith, K.S. and J.Marshall, 2009 Evidence for enhanced eddy mixing at middepth in the Southern Ocean, *J. Phys. Oceanogr.*, **39**, 50-69

WOCE Data Products Committee, 2002, WOCE Global Data, Version 3.0, WOCE International Project Office, WOCE Report No. 180/02, Southampton, UK.

[Figure]

**Fig.1.** Borrowed from Canuto et al. (2017a). Comparison of $|\mathbf{u}_d|$ derived in D3 and CD5 with the data of Fu (2009) and Chelton and Schlax (2011) at $150^0\,$W and $110^0\,$W. The data are reproduced satisfactorily. In all the figures, the model results were obtained from an average of the last 3 years of a simulation with the GISS ER stand-alone OGCM which was run for 300 years.

[Figure]

**Fig.2.** Borrowed from Canuto et al. (2017b). Comparison of the z-profile of the EKE derived in d3 and CD5 in units of its surface value vs. WOCE data in different locations. The model results reproduce the data satisfactorily.

[Figure]

**Fig.3** Borrowed from Canuto et al. (2017b). Comparison of mesoscale diffusivity (in $m^2s^{-1}$) computed within D3 and CD5 model vs. the measured data of Philips and Rintoul (2000, PR00) in the ACC (143E, 51S). The $\varpi = 1$ case is with the contribution of corrections of the higher order in the small parameter equal to the ratio (mean K/EKE).

---

## Author Comment (AC4) · 21 Sep 2017

Remark to the reply to the comments of R2

I found only the paper by Bruggemann and Eden (2015) which presents the spectra of the different works contributing to the EKE production spectrum. The spectrum of the advection contribution which is the sum of that of the mean velocity and the NL force (transfer), is presented in their Figs. 3b,d for different mean Richardson numbers (solid lines). The spectra are the largest (in modulus) and negative at the scale $\sim$Rd and change the sign at larger scales in agreement with the D3/CD5 model.